# Organoboron Ionic Liquids as Extractants for Distillation Process of Binary Ethanol + Water Mixtures

**Ilsiya M. Davletbaeva [1],\*, Alexander V. Klinov [2],\*, Alina R. Khairullina [2], Alexander V. Malygin [2], Sergey E. Dulmaev [1], Alisa R. Davletbaeva [1] and Timur A. Mukhametzyanov [3]** 

[1] Technology of Synthetic Rubber Department, Kazan National Research Technological University, 68 Karl Marx st., Kazan 420015, Russia; impsble@gmail.com (S.E.D.); ilyt25@mail.ru (A.R.D.)

[2] Department of Chemical Process Engineering, Kazan National Research Technological University, 68 Karl Marx st., Kazan 420015, Russia; apelsinochka91@mail.ru (A.R.K.); mav@kstu.ru (A.V.M.)

[3] Department of Physical Chemistry 18 Kremlyovskaya st., Kazan Federal University, Kazan 420008, Russia; timur.mukhametzyanov@kpfu.ru

\* Correspondence: davletbaeva09@mail.ru (I.M.D.); alklin@kstu.ru (A.V.K.)

**Abstract:** Aminoethers of boric acid, which are organoboron ionic liquids, were synthesized by using boric acid, triethanolamine, and triethylene glycol/diethylene glycol. Due to the formation of intermolecular complexes of borates, the structure of aminoethers of boric acid contains ion pairs separated in space, giving these compounds the properties inherent to ionic liquids. It is established that the thermal stability of aminoethers under normal atmospheric conditions increases with an increase in the size of the glycol. According to measurements of fast scanning calorimetry, density, dynamic viscosity, and electrical conductivity, water is involved in the structural organization of aminoethers of boric acid. The impact of the most thermostable organoboron ionic liquids on the phase equilibrium conditions of the vapor–liquid azeotropic ethanol–water mixture is studied. It is shown that the presence of these substances leads to increase in the relative volatility of ethanol. In general, the magnitude of this effect is at the level shown by imidazole ionic liquids, which provide high selectivity in the separation of aqueous alcohol solutions. A large separation factor, high resistance to thermal oxidative degradation processes, accompanied by low cost start reagents, make aminoethers of boric acid on the basis of triethylene glycol a potentially effective extractant for the extractive distillation of water–alcohol mixtures.

**Keywords:** extraction; ionic liquids; vapor–liquid equilibrium; aqueous solution; physicochemical properties

## 1. Introduction

Ionic liquids (ILs) are a new class of compounds consisting only of bulk cations and anions. Particular interest in ionic liquids is related to their unique physical, physicochemical, electrochemical, plasticizing properties [1]. Specific properties include low melting point (<100 °C), practical absence of saturated vapor pressure, good polarity, good dissolving ability, possibility of regeneration, and incombustibility. Currently, work is underway to reduce toxicity of ionic liquids [1–4].

ILs are characterized by a variety of structures. The ability to alter the nature of cations and anions allows adjusting the chemical and physical properties. As a result, one can achieve the required properties by choosing a certain combination of cations and anions from the well-known dependences between the properties and structure of ions in ILs [5–8]. The ability to control the properties of ILs allows one to replace traditional organic solvents and to use them in various fields. For example,

ILs are successfully used as effective solvents [9–14], reactionary and catalytic zones [15,16], and electrolytes [17–20].

From the point of view of practical use of ILs as solvents, their behavior in contact with water, which is determined by the nature of the ILs itself, is important. Thus, a great number of halogen-containing ILs are miscible with water, and some are not miscible at all. Most ILs contain a small amount of residual water acquired by them during the synthesis process. In addition, many ILs are hygroscopic [21,22].

A promising area of ILs practical use is their application as extracting agents for the separation of azeotropic or closely boiling liquid mixtures [23–25]. The main factor inhibiting work in this direction is their high cost, as well as insufficient knowledge of the thermophysical properties of their solutions. Thermophysical properties are necessary to model the development of methods for the regeneration of ILs from solutions, searching for effective options for their practical application, for example, in the processes of substances separation [26].

Extractive distillation has several advantages over traditional separation technologies: it is operated like a conventional distillation process, using two key variables such as polarity and boiling point difference, and, except for the solvent recovery operation, it does not require additional operations to purify products [27–29].

In this work, the objects of the study are organoboron ionic liquids—aminoethers of boric acid (AEBA–TEG/AEBA–DEG), based on boric acid, triethanolamine (TEA) and triethylene glycol (TEG)/diethylene glycol (DEG), which form intermolecular complexes due to the formation of borates (Figure 1). As a result, the structure of aminoethers of boric acid (AEBA) contains ion pairs that are separated in space.

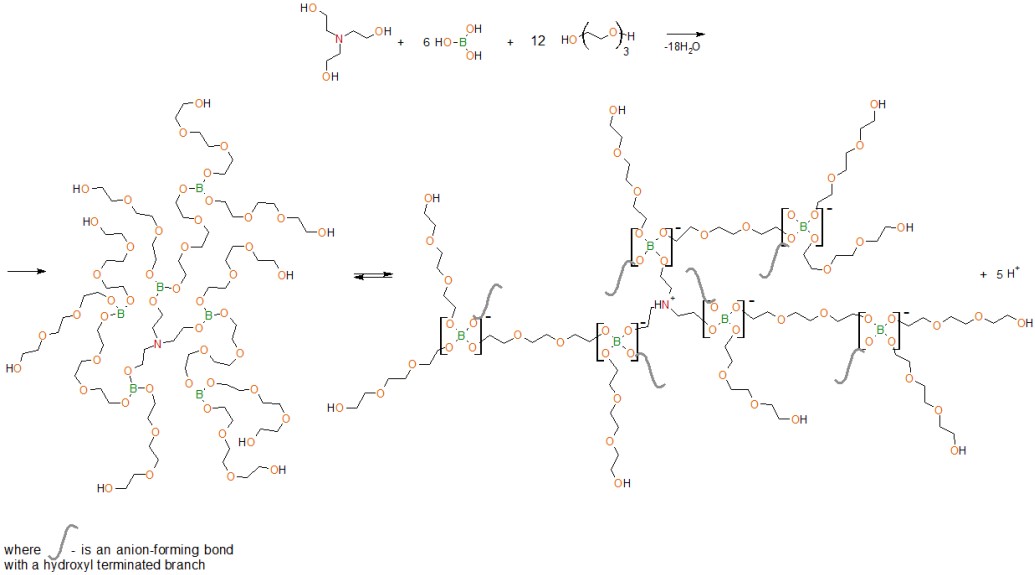

**Figure 1.** Scheme of synthesis of aminoethers of boric acid based on triethylene glycol (AEBA–TEG) and subsequent intermolecular complexation.

In previous works [30–32], by the NMR spectroscopy [31], it was proved that boron atoms in AEBA exist in different forms. The presence of spatial ionic pairs in AEBA resulted there properties inherent in ILs. The size distribution of AEBA was studied [31], a unimodal narrow particle size distribution is observed, indicating the formation of clusters of the same size. It was also concluded [30] that proton concentration released during the complexation of AEBA is inversed to molecular weight of glycol, which is used in AEBA synthesis. Water addition up to 1%–2% leads to a significant decrease in specific resistance. It is explained by the presence of small molecules of water in the system, which facilitates proton transfer under the conditions of a superimposed electric field. The pattern in specific resistance change confirms the accuracy of the statements of AEBA intermolecular complex formation

synthesized with low-weight glycols. This detail allows the classification of this material as an ionic liquid with proton conductivity.

The aim of this work is to study organoboron ionic liquids based on aminoethers of boric acid as extractants for distillation process of binary mixtures.

## 2. Materials and Methods

### 2.1. Materials

All glycols, i.e., diethylene glycol (DEG), triethylene glycol (TEG), and monoethylene glycol (MEG), were purchased from PJSC Nizhnekamskneftekhim (Nizhnekamsk, Russia). Triethanolamine (TEA) was purchased from OJSC Kazanorgsintez. Boric acid (99.99%) was purchased from Sigma-Aldrich. Glycols were additionally dehydrated at a vacuum depth of 1–3 mm Hg and a temperature of 90 °C to a moisture content less than 0.01 wt%. These conditions were chosen experimentally and they ensure the constancy of the residual water content.

### 2.2. Synthesis Process

Aminoethers of boric acid based on mono-, di- and tri-ethylene glycol (AEBA–MEG/AEBA–DEG/AEBA–TEG) were obtained in one-step. The calculated amount of triethanolamine, boric acid and MEG/DEG/TEG was added to a three-necked round bottom flask at a molar ratio of [TEA]:[H3BO3]:[MEG/DEG/TEG] = 1:6:12. The use of such molar ratio was justified in previous works [30–32]. The mass of boric acid (6 mol) was 2.793 g, triethanolamine (1 mol) was 1.124 g. The mass of MEG/DEG/TEG (12 mol) was 5.641 g/9.644 g/13.485 g, respectively. The reaction mixture was heated to 90 °C with heating rate of 2 °C·min$^{-1}$ at a residual pressure of 10 mm Hg and was kept under these conditions for 2 h. This time has been chosen experimentally and it ensures the constancy of the residual water content involved in the structural organization of AEBA. The vacuum was created with oil pump, connected to a U-shaped moisture trap filled with zeolite. Since the dissolution rate is greater than the reaction rate, the mixing was carried out by the natural bubbling of water released during the reaction.

The progress of the reaction was monitored by titration by determining the concentration of hydroxyl groups. When the number of hydroxyl groups reached the plateau, the synthesis was considered as completed. The synthesis product was placed into sealed jar. AEBA–TEG contained 3.96 wt% of water, AEBA–DEG contained 4.97 wt% of water, and AEBA–MEG 5.21 wt% of water. The water content was measured using a volumetric titrator from Mettler Toledo V20 according to the Karl Fischer method.

### 2.3. Preparation of Aqueous Solutions

For the preparation of AEBA aqueous solutions, deionized water was used. Samples were prepared on a ShincoADJ scales with a measurement error of ±0.0001 g.

### 2.4. Viscometry and Determination of Density

The dynamic viscosity of the samples was determined in the temperature range from 22 °C to 100 °C at atmospheric pressure on an SVM 3000 Stabinger Viscometer (Anton Paar, Graz, Austria), with a systematic error of ±0.35% of the measured value. At the same time, the density of the samples was determined with a systematic error of 0.0005 g/cm$^3$.

### 2.5. Determination of Electrical Conductivity of Solutions

The conductivities of aqueous solutions of 3.96–99.95% AEBA–TEG and AEBA–DEG were measured using a Crison GLP 31+ conductivity meter with a measurement error of ±0.5% at 20 °C.

### 2.6. Thermogravimetric Analysis (TGA) Combined with Fourier-IR and Exhaust Gas Mass-Spectroscopy

Samples were analyzed on an Simultaneous Thermal Analyzer 6000 (PerkinElmer, Waltham, MA, USA), in the range of 30–500 °C at a rate of 10 deg/min in an oxidizing air atmosphere. Gas-phase decomposition products of samples were analyzed using a Frontier (PerkinElmer) gas cell of IR Fourier spectrometer, scanning range 4000–500 cm$^{-1}$; resolution 4 cm$^{-1}$) and mass spectroscopy on a Clarus 680 gas chromatograph with an SQ 8C mass spectrometric detector. The gas was taken using a transfer line at temperatures of maximum destruction of the samples.

### 2.7. Fast Scanning Calorimetry Measurement

The fast scanning calorimetry (FSC) measurement was carried out using FlashDSC1 device by Mettler Toledo (Switzerland, Zürich) [33]. The fast scanning calorimeter uses Multistar UFS1 calorimetric sensor with an active diameter of 500 μm [34]. Before the experiment, the sensor was conditioned and corrected according to the manufacturer's instructions to ensure proper relation between the measured signal and the temperature of the sample. The instrument is outfitted with an intracooler, which permits measurements from −90 °C. The measurements were performed under a dynamic nitrogen atmosphere at 30 mL/min flow rate. A thin copper wire (30 μm) was used to place the droplet of the sample in the center of the calorimetric chip.

### 2.8. Phase Equilibrium Experiments

To accurately determine the vapor–liquid phase equilibrium conditions, the Rose–Williams still method and its modifications [35,36] are currently used, which became a further development of the open evaporation method, also known as simple distillation or Rayleigh distillation. Open evaporation is a periodic distillation with one equilibrium stage, in which the generated vapor is continuously removed, so the vapor is in equilibrium with the liquid of the stationary tank at any moment. The method of open evaporation does not allow to obtain the exact concentration of equilibrium phases at the point of interest in the diagram, like the Rose–Williams still method, but it allows to obtain curves of evaporation residues and distillation lines that are strictly determined by the vapor–liquid phase equilibrium conditions of the system. The method of open evaporation is used to study the phase equilibrium of azeotropic mixtures to obtain residue curves, which are then used to determine the lower and upper product of the distillation column, distillation boundary lines, and distillation regions [37]. The open evaporation method, compared to the Rose–Williams still method, is less laborious and less demanding on the experimental conditions, but at the same time provides a qualitative and quantitative assessment of the addition effect of eutectic solvents on the phase equilibrium conditions in the azeotropic mixture in a certain concentration range. Thus, an open evaporation method was chosen to conduct a comparative study of the effect of various AEBAs on the equilibrium conditions of an azeotropic ethanol–water mixture.

To conduct experimental studies on the phase equilibrium of vapor–liquid in the ethanol–water–AEBA system, an IKA-RV 10 (IKA, Germany, Staufen im Breisgau) digital rotary evaporator was used. The initial liquid mixture of a given composition in an amount of 200–250 g was poured into a cube-evaporator, which was immersed in an oil bath. To ensure intense boiling, the temperature fluid in an oil bath was set 10÷20 °C higher than the boiling temperature of the mixture, which was determined in preliminary experiments. The distillate was collected in a receiving flask and, after accumulation it in an amount of 10 ÷ 20 g, was taken for analysis of the composition. The receiving flask was replaced. The evaporation process was carried out continuously with sequential selection of 6–8 portions of distillate, to a residual content of 1/10 part of the amount of the initial mixture in the evaporator cube. The cube residue was weighed and its component composition was determined. The data obtained were checked for compliance with the material balance; the error in the material balance did not exceed 1.5%. The obtained distillates are a binary mixture of ethanol–water, therefore,

in this situation, to determine the water content was used a Mettler Toledo V20 volumetric titrator according to the Karl Fischer Method.

## 3. Results

### 3.1. Thermal Behavior of Aminoethers of Boric Acid

Since processes of an extractive distillation of water–alcohol mixtures were studied at temperatures exceeding 100 °C, it was necessary to determine thermo-oxidative stability of AEBA.

According to Figure 2 and Table 1, the nature of the glycol used has a significant effect on the thermo-oxidative stability of the resulting AEBA. Thus, AEBA–TEG retains its original structure up to T = 200 °C. Despite the fact that AEBA–TEG contains 3.96 wt% of water, and AEBA–DEG contains 4.97 wt% of water, on the TGA curves there is no noticeable weight loss at 100 °C. The loss of 5% of the mass for AEBA–DEG is achieved at T = 149 °C, and for AEBA–TEG it is achieved only at T = 175 °C. The observed patterns may be due to the fact that solvated water is part of the structural organization of AEBA–TEG and AEBA–DEG. According to mass spectroscopic studies of the products released at T = 245 °C during thermo-oxidative degradation of AEBA–TEG (Figure 3), dissociation of AEBA–TEG is not the main process accompanying its thermal decomposition in air. Thermo-oxidative degradation is accompanied by the formation of TEG, crown ether (15-crown-5) and boratrane.

A decrease in length of glycol component in the AEBA structure by replacing TEG with DEG leads to a noticeable decrease in stability to the thermo-oxidative action of corresponding AEBA–DEG (Figure 2, Table 1). According to mass-spectroscopic studies (Figure 3), thermo-oxidative degradation products of AEBA–DEG consist of DEG and boratrane.

Further decreasing of glycol components length to its monomeric form leads to an unacceptable decrease of thermo-oxidative stability of AEBA–MEG for further studies.

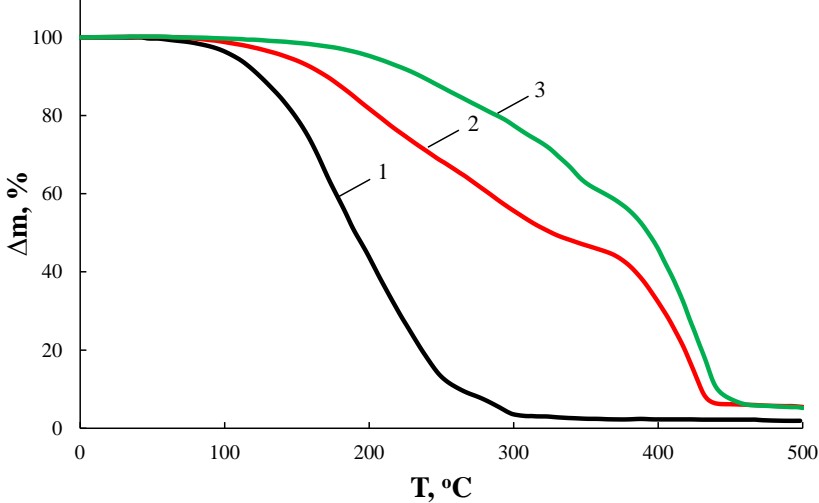

**Figure 2.** TGA curves of AEBA–MEG (**1**), AEBA–DEG (**2**), and AEBA–TEG (**3**) in air.

**Table 1.** Characteristics of thermo-oxidative stability of AEBA-MEG, AEBA-DEG AEBA-TEG for the heating rate of 10 °C/min.

| Sample | $T_{\Delta m\ 5\%}$, °C | $T_{\Delta m\ 10\%}$, °C | $T_{\Delta m\ 50\%}$, °C |
|---|---|---|---|
| AEBA–MEG | <100 | 123 | 190 |
| AEBA–DEG | 149 | 171 | 327 |
| AEBA–TEG | 175 | 233 | 392 |

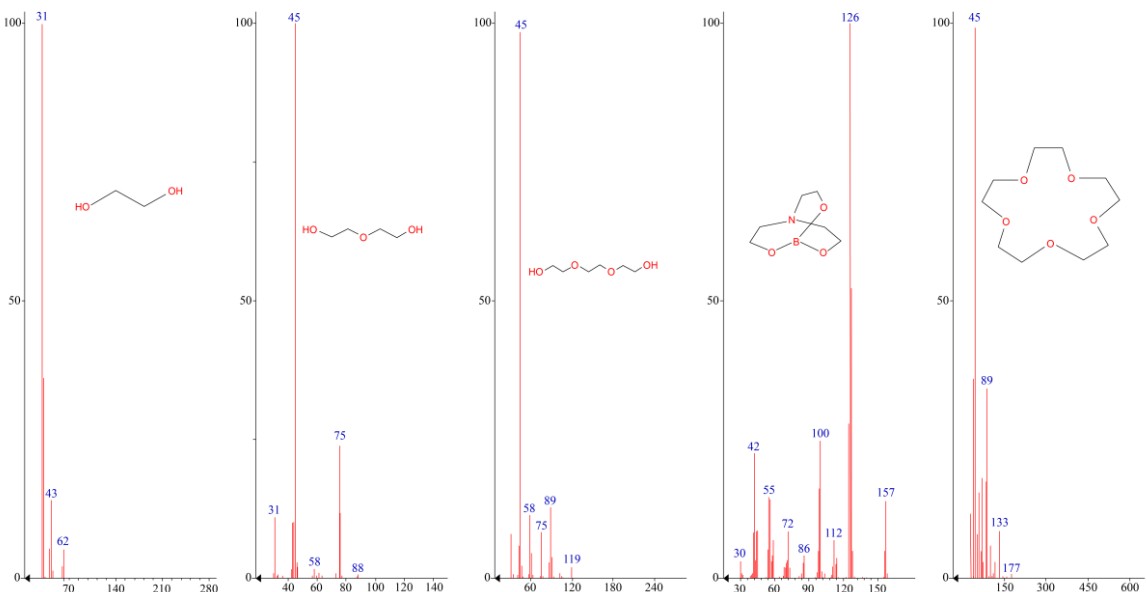

**Figure 3.** Mass-spectra of thermo-oxidative degradation products of AEBA.

The main products of thermo-oxidative degradation here are MEG and boratrane (Figure 3). Boratrane is also formed in decomposition cases of both AEBA–TEG and AEBA–DEG. Additional products of AEBA–TEG decomposition are 15-Crown-5 and triethylene glycol, and in the case of AEBA–DEG diethylene glycol is observed (Figure 3). The nature of the decomposition products of AEBA in terms of flammability and toxicity allows them to be used at temperatures below the start of decomposition. In this way, AEBA–TEG, and AEBA–DEG can be used in processes of extractive distillation of water-alcohol mixtures due to their high resistance to thermo-oxidative degradation processes.

### 3.2. Fast Scanning Calorimetry

Figure 4 presents the results of FSC analysis of AEBA–TEG with a water content of 3.96 wt%. Several successive heating and cooling of the sample was carried out at a rate of 1000 K/s; the graph shows the first and second heating of the sample. The first heating (black lower curve) showed the presence of two endothermic thermal effects: narrow at −60 °C and wide in the range from 0 to 200 °C. Apparently, both of these effects are associated with the presence of water in the test sample, which leaves during the first heating. Noteworthy is the absence of an endothermic effect that could be associated with the melting of water crystals or water–IL eutectic. On the second heating (blue curve) AEBA–TEG has a stable behavior in the temperature range −80 to 240 °C with the presence of a glass transition at −30 °C. The same picture is observed during subsequent heating. A glass transition is also observed on the cooling curve (red curve), and there are no other types of effects. Thus, the wide endothermic effect observed during the first heating can be unambiguously associated with the evaporation of bound water, while the evaporation process ends at a temperature noticeably above 100 °C, which indicates a fairly strong interaction of water with AEBA–TEG. The narrow effect at −60 °C is apparently a glass transition with an additional relaxation effect. A decrease in the transition temperature compared with the anhydrous compound indicates the plasticizing effect of water.

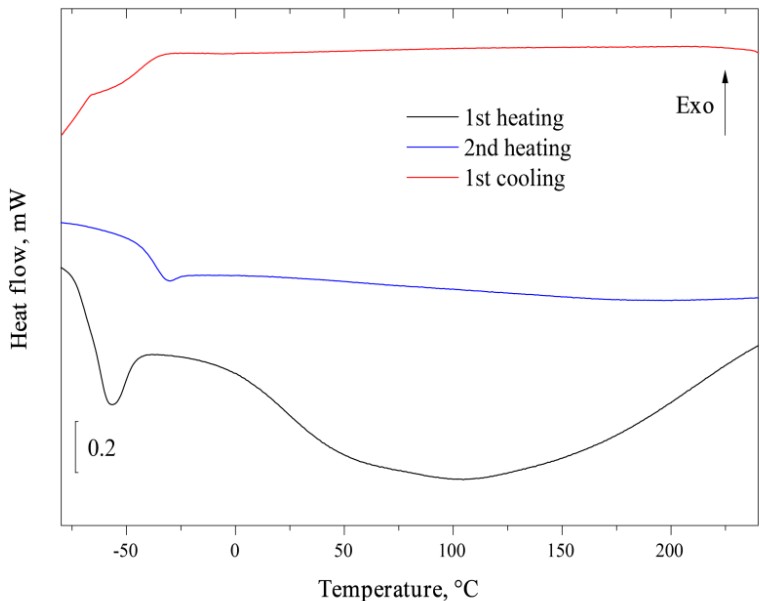

**Figure 4.** Fast scanning calorimetry curves of AEBA–TEG.

### 3.3. The Study of Aqueous Solutions of AEBA–TEG and AEBA–DEG

Since the used ILs contain solvated water, the content of residual water was taken into account when preparing aqueous solutions of AEBA. The dependences of the density, dynamic viscosity, and electrical conductivity of aqueous solutions of AEBA on its content were studied.

The measured dependences in density of AEBA aqueous solutions on temperature and composition are shown in Table 2. It is clear that the densities of AEBA–DEG and AEBA–TEG aqueous solutions are barely different (less than 1 wt%), while increase of AEBA concentration to 90 wt% leads to an increase in the density of aqueous solution of relatively pure water to 20%. The correlation observed between the patterns of change in the shape of electronic spectra and densities of aqueous solutions of AEBA–TEG is most probable result of the involvement of water into structural organization of AEBA–TEG.

**Table 2.** The densities of aqueous solutions of AEBA–TEG and AEBA–DEG.

| Density, g/cm³ | Temperature, °C | | | | | Temperature, °C | | | |
|---|---|---|---|---|---|---|---|---|---|
| | 20 | 40 | 60 | 80 | 100 | 20 | 40 | 60 | 80 |
| AEBA, wt% | AEBA–TEG | | | | | AEBA–DEG | | | |
| 0 | 0.998 | 0.992 | 0.983 | 0.971 | - | 0.998 | 0.992 | 0.983 | 0.971 |
| 10 | 1.020 | 1.012 | 1.001 | 0.99 | - | 1.022 | 1.014 | 1.005 | 0.992 |
| 20 | 1.043 | 1.033 | 1.022 | 1.009 | - | 1.048 | 1.032 | 1.028 | 1.015 |
| 30 | 1.065 | 1.054 | 1.042 | 1.029 | - | 1.073 | 1.062 | 1.05 | 1.036 |
| 40 | 1.091 | 1.079 | 1.065 | 1.050 | 1.034 | 1.100 | 1.088 | 1.075 | 1.059 |
| 50 | 1.115 | 1.101 | 1.087 | 1.071 | 1.054 | 1.127 | 1.113 | 1.099 | 1.083 |
| 60 | 1.137 | 1.122 | 1.107 | 1.091 | 1.074 | 1.151 | 1.136 | 1.122 | 1.106 |
| 70 | 1.157 | 1.142 | 1.126 | 1.110 | 1.093 | 1.174 | 1.159 | 1.144 | 1.128 |
| 80 | 1.175 | 1.160 | 1.144 | 1.127 | 1.110 | 1.194 | 1.180 | 1.165 | 1.148 |
| 85 | 1.181 | 1.166 | 1.149 | 1.133 | 1.116 | - | - | - | - |
| 86.5 | - | - | - | - | - | 1.206 | 1.191 | 1.175 | 1.158 |
| 90 | 1.187 | 1.171 | 1.155 | 1.139 | 1.122 | - | - | - | - |
| 91 | 1.189 | 1.172 | 1.156 | 1.140 | 1.123 | - | - | - | - |
| 92 | 1.190 | 1.173 | 1.157 | 1.140 | 1.123 | - | - | - | - |
| 93 | 1.190 | 1.174 | 1.157 | 1.140 | 1.123 | - | - | - | - |
| 94 | 1.191 | 1.175 | 1.158 | 1.141 | 1.125 | - | - | - | - |
| 95.4 | 1.191 | 1.174 | 1.157 | 1.141 | 1.124 | - | - | - | - |

Figure 5 shows the values of dynamic viscosity of aqueous solutions of AEBA. According to these values, aqueous solutions of AEBA–DEG and AEBA–TEG with concentrations of less than 60 wt% possess similar viscosity. However, with a further increase of AEBA content, the viscosity of AEBA–DEG solution grows faster than AEBA–TEG solution. For example, for a concentration of 85 wt% the viscosity of AEBA–DEG solution is 2.5 times higher than that of AEBA–TEG solution. It should be also noted that the temperature dependence of the viscosity of AEBA aqueous solutions is more significant as compared to pure water and increases with the increase of AEBA concentration. Thus, in the case of water, the viscosity in temperature range from 10 to 80 °C changes threefold, and in the case of 80 wt% AEBA solution the change of viscosity reaches 15–20 times.

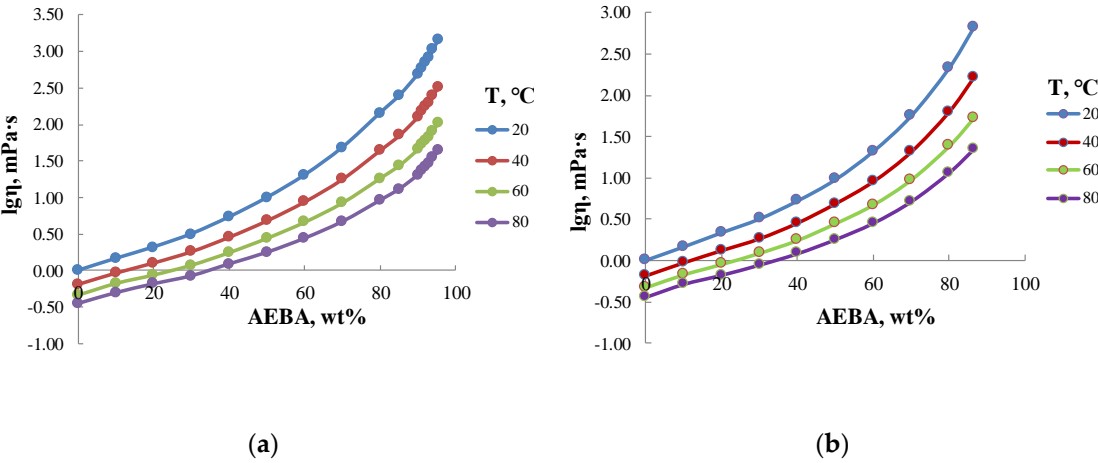

(**a**)  (**b**)

**Figure 5.** Dynamic viscosity of aqueous solutions of AEBA–TEG (**a**) and AEBA–DEG (**b**).

Thus, measurements of the concentration dependences of dynamic viscosity confirm that the formation of borates leads to the formation of spacious intermolecular complexes, while borate-anions themselves engage in dipole-dipole interactions with water. Solvation, in its turn, has a significant impact on the structural organization of AEBA–TEG and AEBA–DEG, and, accordingly, on their physicochemical properties. An additional confirmation of these results were the dependences investigated by the method of electrical conductivity, which is an important approach in studying the properties of ionic liquids [38,39].

Non-additive dependence can be also observed in electrical conductivity measurements of AEBA–DEG and AEBA–TEG aqueous solutions. (Figure 6). Deionized water with a specific electrical conductivity of $\sigma = 1.03$ µS/cm was used for studies. For calculations of molar concentrations of AEBA–DEG, the used molar mass was 1734 g/mol, and for AEBA–TEG, it was 1998 g/mol. As can be seen from Figure 6, aqueous solution of AEBA demonstrates classic properties of concentrated saline solutions with a peak of conductivity, which is inherent for ionic liquids as well. However, in the area of the maximum the conductivity of AEBA aqueous solution is two orders of magnitude lower than, for instance, for imidazole ionic liquids [40]. At high concentrations of AEBA–DEG, the nature of changes in electrical conductivity noticeably differs from more dilute solutions. That is, the solvation of space-separated ionic pairs makes the structure denser and reduces the mobility of charge carriers.

An increase in the molecule size from AEBA–DEG to AEBA–TEG leads to sequential decrease in the electrical conductivity of aqueous solutions of AEBA–TEG. This feature can be explained by the fact, that the increase of the glycol component size in AEBA influences negatively on the formation of intermolecular complexes because of steric hindrances, which reduce the number of protons released and obstruct their migration.

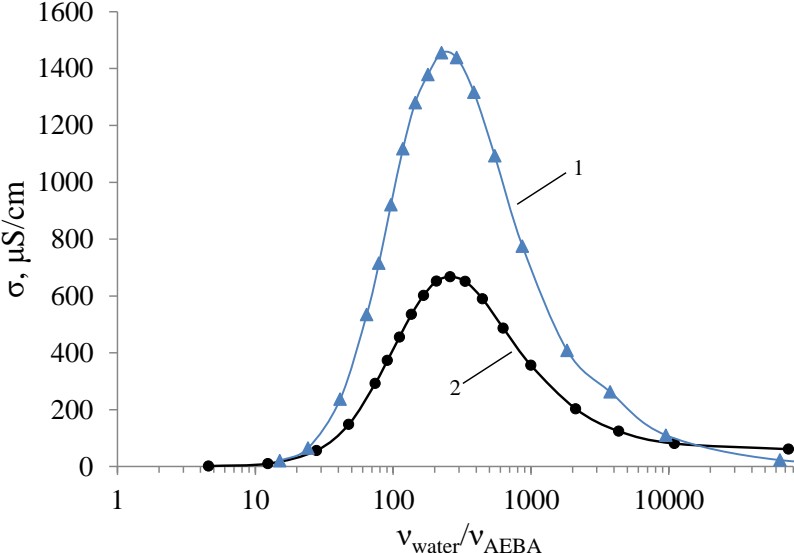

**Figure 6.** Dependences of the electrical conductivity of AEBA–DEG (**1**) and AEBA–TEG (**2**) on their concentration in aqueous solutions (T = 25 °C).

### 3.4. Vapor–Liquid Equilibrium for Ethanol–Water Mixture in the Presence of AEBA

To analyze conditions of phase equilibrium of vapor–liquid according to experimental data obtained in a rotary evaporator, the equation of the process of open evaporation (Rayleigh equations) of a binary mixture was used [37]:

$$y(x)^* = \frac{dx}{d\ln(L)} + x \tag{1}$$

where $x$ and $L$ are composition and mass of boiling mixture; $y(x)^*$ is equilibrium composition of the vapor.

The masses of $i$-th portion of distillate $P_i$ and its composition $y_i$ measured during the experiment allows to determine the dependence of change in composition of the boiling mixture on its mass according to the equation of material balance:

$$x_{i+1} = \frac{L_i x_i - P_{i+1} y_{i+1}}{L_{yi} - P_{i+1}}; \; i = 0 \ldots n-1 \tag{2}$$

where $n$ is the number of samples taken. Based on these data, it is possible to construct the so-called residue curves, which describe the change in the liquid composition of the mixture.

The discrete data obtained according to (2) were approximated by a polynomial in the form of dependence of $x = f(e)$, where $x$ are weight fraction of ethanol in the boiling mixture; $e = P/L_0$ is distillate rate; $L_0$ is the initial mass of mixture, which was in the cube evaporator. Next, substituting this dependence into (1), the equilibrium composition of the vapor can be calculated as follows:

$$y(x)^* = \frac{dx}{de}(1-e) + x \tag{3}$$

The error in the equilibrium compositions of vapor and liquid, determined in this way, is associated with the presence of additional processes of partial condensation (evaporation), i.e., with stages of separation, which can be in the tube part of flask on its way from edge of boiling liquid to the vapor condensation area.

The magnitude of this discrepancy would depend on relative volatility of the components for the given concentration of the solution. To estimate the error in determining the equilibrium concentration, distillation experiments were conducted on binary mixture of ethanol–water at atmospheric pressure. The results of the comparison are shown in Figure 7. The calculated data were obtained via solving

the Equation (3), in which the dependence of equilibrium vapor composition on the composition of liquid phase was determined based on the non-random two liquid (NRTL) model [41]. The NRTL model was chosen because it describes well the vapor–liquid equilibrium conditions for non-ideal solutions. The parameters of this model for the ethanol–water system are known in the literature, which provide high accuracy in calculating the conditions of vapor–liquid equilibrium at various pressures and temperatures [42,43]. According to the figure provided, the experimental data agree well with the calculated ones in cases, where initial concentration of ethanol $x_0$ in the mixture, which was in the cube-evaporator, is higher than 0.60 wt.fr. (for 0.84 wt.fr. of $x_0$, the maximum error was 0.35%; for 0.65 wt.fr. of $x_0$—3.8%; for 0.5 wt.fr. of $x_0$—29%), and where relative volatility of the components is low. At lower concentrations the error increases, which is explained by the presence of additional stages of separation in the rotatory evaporator. Since the number of separation stages under equal conditions of the experiment in the rotary evaporator should not change, it is possible to estimate the influence of AEBA on conditions of the vapor–liquid phase equilibrium by comparing the results of the distillation experiment of a binary and tri-component mixture. In addition, this measurement method is less time-consuming, in comparison with using the Rose–Williams still, and in one experiment, it allowed to obtain data in the range of concentration measurements.

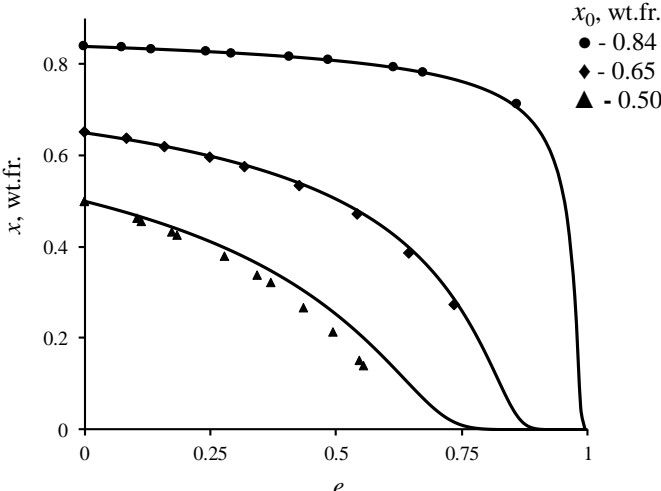

**Figure 7.** Change in the concentration of ethanol in the ethanol–water liquid mixture during open evaporation: line is solution (3); geometric figures—experimental data.

Figure 8 shows the results of such a comparison. The lines show the distillation of a binary ethanol–water mixture at atmospheric pressure, the geometric figures show the results of the distillation of these mixtures in the presence of AEBA. As can be seen from the curves, the addition of AEBA increases the relative volatility of ethanol, which leads to more intensive depletion in the cube. For quantitative estimation of the AEBA impact, the following ratio was used, determining the change in relative volatility of ethanol:

$$m = \frac{y_1^{AEBA}}{y_1^{exp}}$$

where $y_1^{AEBA}$ and $y_1^{exp}$ are the experimentally determined equilibrium concentrations of ethanol in vapor over tri-component and binary solution at equal amount of ethanol and water. AEBA in comparison with water and ethanol can be considered as non-volatile component, so its concentration in vapor equals 0. Next, it is possible to estimate the coefficient of relative volatility:

$$\alpha_{12} = \frac{m y_1^0 / x_1}{\left(1 - m y_1^0\right)/x_2} \tag{4}$$

where $y^0$ is equilibrium concentration of ethanol in the vapor over binary solution determined by the Non-Random Two-Liquid model (NRTL); $x_1$ and $x_2$ are concentrations of ethanol and water in the solution.

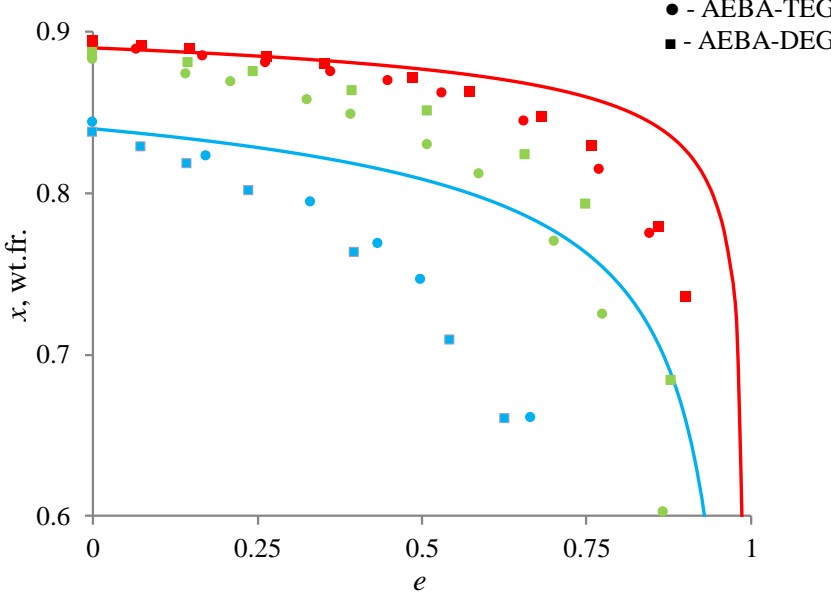

**Figure 8.** Change in the concentration of ethanol in a liquid mixture during open evaporation: red color is 0.1 wt.fr. of AEBA; green—0.2 wt.fr.; blue—0.6 wt.fr.

The results of all experiments on investigation of the impact of AEBA on vapor–liquid phase equilibrium are presented in Table 3.

**Table 3.** Coefficient of relative volatility of ethanol depending on AEBA–DEG/AEBA–TEG concentration.

| $w^{AEBA}$ | Interval of Liquid Phase Compositions Based on Ethanol $x$ | $\alpha_{12}$(AEBA–TEG) | $\alpha_{12}$(AEBA–DEG) |
|---|---|---|---|
| 0.25 | 0.84–0.87 | 1.95 | 1.65 |
| 0.30 | 0.82–0.85 | 2.21 | 1.80 |
| 0.40 | 0.78–0.86 | 2.89 | 2.05 |
| 0.50 | 0.64–0.81 | 3.37 | 2.50 |
| 0.60 | 0.53–0.83 | 4,17 | 4.50 |
| 0.70 | 0.37–0.77 | 4.50 | 4.95 |
| 0.75 | 0.59–0.73 | 5.04 | 5.68 |

$w^{AEBA}$ is mass fraction of AEBA in the three-component mixture; $x$ is mass fraction of ethanol in the absence of ionic liquid.

As can be seen from Table 3, with the increase of AEBA concentration, the coefficient of relative volatility of ethanol increases, including the area close to the azeotropic point. The inclination angle of dependence of the relative volatility of ethanol on AEBA concentration for AEBA–TEG is different compared to the case of AEBA–DEG. This leads to the situation where at low concentrations of AEBA, the relative volatility of ethanol in the presence of AEBA–TEG is greater than in the presence of AEBA–DEG. At higher concentrations ($w^{AEBA} > 0.6$), the situation is reversed.

For evaluation of AEBA efficiency, the relative volatility coefficient of ethanol was compared with published data, in which vapor–liquid phase equilibrium in the ethanol–water system was studied in the presence of imidazole ionic liquids [44–48]. According to the analysis, the values of relative volatility coefficients of ethanol depend on the nature of cation and anion of ionic liquid. The greatest

effect of increasing the relative volatility of ethanol near the azeotropic point is provided by ILs with [Cl] and [OAc] anions [45]. With the increase in the length of alkyl chain of the cation, the coefficient of relative volatility decreased. For the most effective ionic liquid [Emim][Cl], the value or relative volatility of ethanol at $x = 0.95$ for w[Emim][Cl] = 0.2 is $\alpha_{12} = 1.4$, and for w[Emim][Cl] = 0.6, $\alpha_{12} = 3.4$. Thus, the separation ability of AEBA–TEG/AEBA–DEG for ethanol–water mixture corresponds to imidazole ionic liquids with the greatest impact on relative volatility of ethanol. At the same time, the simplicity of AEBA synthesis, low cost, and its high stability under ordinary atmospheric conditions create advantages for their practical usage as a potentially effective reagent for processes of extractive distillation of ethanol–water mixtures.

## 4. Conclusions

The effect of glycol components length on thermo-oxidative stability of organoboron ionic liquids was studied. It was shown that resistance to thermo-oxidative degradation of AEBA decreases with a decrease of glycol components length.

The influence of concentration of AEBA–TEG and AEBA–DEG on the density, dynamic viscosity, and electrical conductivity was studied. AEBA–TEG has also been studied using fast scanning calorimetry. It was established that AEBA exist in the solvated state, i.e., water is an integral part of their structural organization.

In connection with the results obtained, AEBA–DEG and AEBA–TEG provided enough stability to be used as extractants in extractive distillation process. According to the separation experiment results, regarding the relative volatility coefficient of ethanol, it was found that the presence of AEBA–TEG in the initial mixture could greatly affect this value. The advantages of the usage of AEBA–TEG as an extracting agent for the separation of mixtures based on the example of azeotropic ethanol–water mixture was shown and compared to known ionic liquids.

**Author Contributions:** I.M.D. designed the study, coordinated the study, carried out data and results analysis, carried out sequence alignments, and drafted the manuscript; A.V.K. conceived of the study, coordinated experimental part of the study, carried out the results analysis, and critically revised the manuscript; A.R.K. carried out separation experiment, determined electrical conductivity, density and viscosity of aqueous solutions, and collected data; A.V.M. coordinated phase equilibrium experiments, carried out sequences of the study; S.E.D. carried out synthesis lab work and participated in the manuscript drafting; A.R.D. carried out synthesis lab work, participated in TGA analysis; T.A.M. carried out fast scanning calorimetry experiment and analyzed the obtained results. All authors have read and agreed to the published version of the manuscript.

**Funding:** This work was supported by the Russian Science Foundation (grant No. 19-19-00136).

**Conflicts of Interest:** There are no conflicts to declare.

## Abbreviations

| | |
|---|---|
| TEA | triethanolamine |
| MEG | monoethylene glycol |
| DEG | diethylene glycol |
| TEG | triethylene glycol |
| AEBA | aminoethers of boric acid |
| AEBA–MEG | aminoethers of boric acid based on monoethylene glycol |
| AEBA–DEG | aminoethers of boric acid based on diethylene glycol |
| AEBA–TEG | aminoethers of boric acid based on triethylene glycol |
| TGA | thermogravimetric analysis |
| $L_0$ | initial mass of mixture |
| $x_0$ | initial concentration of ethanol in the mixture |
| $L$ | mass of boiling mixture |
| $P$ | mass distillate |
| $e$ | distillate rate |
| $\alpha_{12}$ | coefficient of relative volatility |

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
