# Peer review of "Organoboron Ionic Liquids as Extractants for Distillation Process of Binary Ethanol + Water Mixtures"

_processes, doi:10.3390/pr8050628_

Round 1

Reviewer 1 Report

The authors carried out the VLE study on the effect of organoboron ionic systems on the distillation for the ethanol-water mixture.

It should be noted that the term Ionic Liquids corresponds to ionic compounds the composition of which is predefined. It is not applied to the systems where the composition and molar mass is not totally clear. The system studied in the current work corresponds to the systems known as Deep Eutectic Solvents (DES).  In this system, the correct composition of the studied material should be provided: molar mass range, the conditions of each acidic center, the precise amount of each solvent after the purification procedure.

 Authors widely used abbreviations of the studied compounds but in the case of abstract, the abbreviations should be removed. The abstract will be read separately from the main article.

The other comments are given in the attached manuscript file.

Reviewer 2 Report

Aminoethers of boric acid (AEBA-TEG / AEBA-DEG), which are organoboron ionic liquids, were synthesized by using boric acid, triethanolamine and triethylene glycol (TEG) /diethylene glycol (DEG). Due to the formation of intermolecular complexes of borates, the structure of aminoethers of boric acid (AEBA) contains ion pairs separated in space, giving these compounds the properties inherent to ionic liquids. It is established that the thermal stability of AEBA under normal atmospheric conditions increases with an increase in the size of the glycol. According to measurements of fast scanning calorimetry, density, dynamic viscosity and electrical conductivity,
water is involved in the structural organization of AEBA. The impact of the most thermostable organoboron ionic liquids on the phase equilibrium conditions of the vapor-liquid azeotropic
ethanol-water mixture is studied. It is shown that the presence of AEBA-DEG and AEBA-TEG.

Comments:

Line 33: Most of ionic liquids are also non-toxic [1-8] ¡excess references!...Delete 4 references

Line 39: What do they mean by “effective solvents”?

Line 40: Delete “etc.”

Lines 83 and 84: Glycols were additionally dehydrated at a vacuum depth of 1-3 mm Hg and a 83 temperature of 90 °C to a moisture content less than 0.01%. Why at these conditions?

Lines 87 – 89: The calculated amount of triethanolamine, boric acid and MEG / DEG / TEG was added to a three-necked round bottom flask at a molar ratio of TEA]:[H3BO3]:[MEG / DEG / TEG] = 1:6:12. How did they get to those results?

Line 92: why 2 hours?

Line 289 and 290: Delete “Error! Reference source not found.”

What is the purity of the products obtained?

In the phase equilibrium because they used the NRTL model if there are other models?

Reviewer 3 Report

In this manuscript, the authors report synthesis of inexpensive aminoethers of boric acid, further used in extractive distillation of water-alcohol mixtures. Beside the authors describe useful application of the investigated in their group systems (Ref. 34, 35, 35), there are some inaccuracies  in the manuscript, which in my opinion make it not publishable in the present form.

Some comments and suggestions are as follows.

1) the authors state the nontoxic character of ILs, supporting the claim quite random references 1-8 which do not relate to the toxicity of the ILs. In fact, commonly investigated e.g. imidazolium ILs reveal some toxicity. Recently a paper describing the possibility of its reduction was published (ACS Sustainable Chem. Eng. 2020, 8, 926−938) 

2)Although the organoboron ILs were investigated as extractants for distillation process, there was no explanation for conducting in this study the experiment reporting the electrical conductivities of the systems. 

3)the manuscript lacks description on how the composition analysis during the phase equilibrium experiments was done.

4) fragment from line 184-189 is doubled (192-199), additionally supported by wrong fig. notation (fig. 4 instead of fig.3). References in line 289, 290 are missing.    

Round 2

Reviewer 2 Report

The authors responded well to the review, therefore the article is accepted for publication.

Reviewer 3 Report

All the issues were addressed. I recommend publication of the manuscript in the present form.